# Visual Outcomes and Patient Satisfaction of Enhanced Monofocal Intraocular Lens in Phacovitrectomy for Idiopathic Epiretinal Membrane

**DOI:** 10.3390/bioengineering11090939

**Published:** 2024-09-19

**Authors:** Ji Youn Choi, Yeo Kyoung Won, Soo Jin Lee, Se Woong Kang, Dong Hui Lim

**Affiliations:** 1Department of Ophthalmology, Samsung Medical Center School of Medicine, Sungkyunkwan University, Seoul 06351, Republic of Korea; 2Samsung Eye Clinic, Daegu 41940, Republic of Korea; 3Samsung Advanced Institute for Health Sciences and Technology (SAIHST), Sungkyunkwan University, Seoul 06355, Republic of Korea

**Keywords:** ICB00, ZCB00, phacovitrectomy, epiretinal membrane

## Abstract

Background: To evaluate the clinical outcomes and patient satisfaction after implantation of an enhanced monofocal intraocular lens (TECNIS Eyhance ICB00) in patients with idiopathic epiretinal membrane (ERM) who underwent cataract surgery with pars plana vitrectomy (PPV). Methods: This is a single-center, retrospective, comparative study. In total, 61 eyes of 61 patients with idiopathic ERM and cataracts were included. We measured the uncorrected near and intermediate visual acuity (UNVA and UIVA), uncorrected and corrected distance visual acuity (UDVA and CDVA), central macular thickness, defocus curves, and contrast sensitivity 3–6 months after the surgery. Overall patient satisfaction was assessed using a questionnaire at the 1-month follow-up visit. Results: The ICB00 group showed better near and intermediate visual acuity than the monofocal group (TECNIS ZCB00); however, no statistically significant differences were found between the groups. The ICB00 group exhibited wider defocus curves at near to far distances (−3.0 to +2.0 D) than the ZCB00 group. There were no significant differences in the results of the contrast sensitivity test, dysphotopsia, spectacle dependence, or patient satisfaction between the two groups. Conclusions: In combined PPV and cataract surgery for ERM patients, ICB00 resulted in good visual acuity with a smoother defocus curve compared to the ZCB00 group.

## 1. Introduction

Epiretinal membrane (ERM) is a retinal disorder that provides traction to the inner retinal surface and leads to visual disturbances such as metamorphopsia, macropsia or micropsia, monocular diplopia, and potential visual loss. The global prevalence of the ERM is 2.2–8.5% [1,2]. Idiopathic ERM is particularly common in older individuals, with a prevalence of up to 34% in the population over 60 years of age. Symptomatic ERM with cataract patients undergo combined pars plana vitrectomy (PPV) and cataract surgery [3]. An intraocular lens (IOL) is a crucial factor influencing the quality of vision for patients after cataract surgery, as it replaces the refractive power of the natural crystalline lens. This is no exception for patients undergoing combined PPV and cataract surgery. However, a patient with ERM is generally not an ideal candidate for multifocal IOL due to several complications. These include reduced spherical power predictability, diminished contrast sensitivity, a higher risk of postoperative cystoid macular edema, and less improvement in visual acuity [3]. The literature on the use of extended depth-of-focus or enhanced monofocal IOLs in ERM patients is limited, and there are no established evidence-based guidelines for the choice of IOL in combined PPV and cataract surgery. There are a few studies on patients with retinal diseases who had combined PPV and cataract surgery, which compared the postoperative clinical outcomes of the Eyhance IOL (TECNIS Eyhance ICB00; Johnson & Johnson Surgical Vision, Santa Ana, CA, USA) with those of the monofocal IOL (TECNIS^®^ ZCB00; Johnson & Johnson Surgical Vision) [4,5]. Park et al. reported that the ICB00 group showed better uncorrected intermediate vision without compromising uncorrected and corrected distant vision, as well as contrast sensitivity, compared to the ZCB00 group in patients who underwent combined PPV and cataract surgery [4]. This study included patients with a broad range of retinal diseases such as vitreous hemorrhage, rhegmatogenous retinal detachment, vitreous opacity, epiretinal membrane, macular hole, and vitreomacular traction [4]. Additionally, Hwang et al. reported postoperative improvement in both distant and intermediate vision using the ICB00 in patients with ERM, vitreous hemorrhage, and vitreous opacity who also underwent combined PPV and cataract surgery [5]. These studies suggest that the use of enhanced monofocal IOLs is not contraindicated in eyes with retinal disorders. However, research specifically focused on patients with ERM remains insufficient, despite the relatively high prevalence of ERM among the elderly. Park et al. included 17 patients with ERM, and Hwang et al. included 15 [4,5]. Our study, with a larger number of ERM patients, aimed to investigate the clinical outcomes and patient satisfaction in patients with idiopathic ERM who have undergone combined PPV and cataract surgery.

## 2. Materials and Methods

### 2.1. Patients

This retrospective, single-center, comparative study analyzed 61 eyes of 61 patients with idiopathic ERM and cataracts in the Department of Ophthalmology at Samsung Medical Center between February and June 2022. The study was reviewed and approved by the Institutional Review Board (IRB) of the Samsung Medical Center (# 2023-11-080-001), in accordance with the tenets of the Helsinki Declaration. The requirement for informed consent was waived according to IRB protocols of the Samsung Medical Center (# 2023-11-080-001). Two experienced surgeons (D.H.L. and S.W.K.) collaborated to perform each surgery. D.H.L. is a cataract specialist, and S.W.K. is a retinal specialist. The patients underwent conventional PPV, internal limiting membrane (ILM) peeling, and cataract surgery with posterior capsulotomy. They were divided into two groups based on the type of the implanted IOL: enhanced monofocal IOL (ICB00) and standard monofocal IOL (ZCB00). The inclusion criteria were defined as follows: patients aged 50 to 80 years, preoperative corneal astigmatism ≤1.0 diopter (D) measured using topography, primary ERM diagnosed using optical coherence tomography (OCT) (Spectralis; Heidelberg Engineering, Heidelberg, Germany), and central macular thickness (CMT) under 500 μm. According to the classification of the ERM stage proposed by Hwang and Sohn, all patients in this study were classified as 1B or 1C of Group 1 [6]. Group 1 consisted of ERMs involving the fovea, referred to as the fovea-attached type [6]. Among these, group 1B included ERMs that showed more exaggerated tenting of the outer retinal layer in the foveal region, with slight thickening and distortion of the inner retinal layer due to traction from the ERM [6]. Group 1C involved ERMs characterized by significant inner retinal layer thickening and inward tenting of the outer retinal reflectivity in the foveal zone [6]. The exclusion criteria were defined as follows: patients with corneal irregular astigmatism, treatment-requiring diabetic retinopathy or age-related macular degeneration, any ocular disorders that could significantly affect visual outcomes (e.g., pathologic myopia, keratoconus, corneal opacity or dystrophy, amblyopia, clinically significant dry eye, chronic uveitis, iritis, pseudoexfoliation syndrome, glaucoma, or intraocular pressure >21 mmHg), history of any previous intraocular or corneal surgery, eyes with secondary ERM resulting from uveitis, previous ocular surgery or previous retinal laser treatment, and cases with unexpected intraoperative events which require additional procedures such as intraoperative tamponade (e.g., retinal break or significant vitreous hemorrhage) or significant postoperative complications (e.g., secondary glaucoma or endophthalmitis).

Under retrobulbar or general anesthesia, all surgeries were performed using a three-port 23-gauge vitrectomy system (Constellation Vision System^®^; Alcon Inc., Fort Worth, TX, USA). Trocars were inserted in the superonasal, superotemporal, and inferotemporal quadrant of the bulbar conjunctiva posterior to limbus. Routine phacoemulsification with a clear corneal incision and well-centered in-the-bag IOL implantation was performed. This was followed by sequential core and peripheral vitrectomy, along with epiretinal membrane and ILM peeling using a wide-angle viewing system (Oculus BIOM^®^; OCULUS Surgical, Wetzlar, Germany). At the end of the procedures, the trocars were removed, leakage was checked, and additional sclerotomies were performed if necessary.

### 2.2. Measurements

Preoperatively, optical biometry including axial length, anterior chamber depth, and corneal refractive power were measured by ARGOS^®^ (Alcon Inc., Fort Worth, TX, USA). The target diopter for each patient was determined using SRK/T, Haigis, mainly using the Barrett Universal II formula. Comprehensive ophthalmic evaluations including slit lamp examination, Goldmann applanation tonometry, manifest refraction, and corneal topography were performed. CMT was measured using Spectralis^®^ OCT, and both uncorrected and corrected distance visual acuity (UDVA and CDVA) were evaluated at a distance of 4 m.

The postoperative outcomes were measured at 3–6 months after the surgery. Visual acuity was assessed using the Snellen chart. UDVA and CDVA were evaluated, and uncorrected near visual acuity (UNVA) was measured at the distance of 33 cm and 40 cm. Uncorrected intermediate visual acuity (UIVA) was measured at the distance of 60 cm and 80 cm. Prediction error (PE) was calculated as the difference between the postoperative spherical equivalent (SE) and the target diopter. The mean error (ME) for each group was calculated as the average of all PEs. The mean absolute error (MAE) was measured as the average of the absolute value of each PE, and the median absolute error (MedAE) was measured as the median of the absolute value of each PE. The defocus curves were measured via monocular visual acuity under photopic conditions at a distance of 5 m, adjusting lenses in 0.5 D increments ranging from −4.0 to +2.0 D at 1 month postoperatively. Contrast sensitivity was measured in monocular at 0.5, 1.1, 2.2, 3.4, 7.1, and 14.2 spatial frequencies in cycles per degree (cpd) using Metrovision (MonCv3; Metrovision). These measurements were conducted under mesopic (3 cd/m^2^), mesopic with glare, and photopic (85 cd/m^2^) conditions postoperatively at 1 month. Patient satisfaction was evaluated using a detailed questionnaire on near, intermediate, and far vision; dependence on a spectacle; and overall satisfaction. The patients rated their visual acuity and overall satisfaction with the following five categories: very satisfied, satisfied, neither satisfied nor dissatisfied, dissatisfied, and very dissatisfied at the 1-month postoperative visit. The questionnaire also included visual symptoms (glares, halos, starbursts, hazy or blurred vision, distortion, double vision, fluctuation, focusing difficulty, and difficulty in perception of distance and depth) that may degrade the quality of vision. These visual symptoms were rated for frequency, degree, and bothersomeness on a scale of none, minimal, moderate, and severe (0, 1, 2, and 3 points, respectively).

### 2.3. Statistical Analysis

The Mann–Whitney U-test was used to compare the measurements of the ICB00 group with those of the ZCB00 group. All statistical analyses were carried out using SPSS for Windows (version 21.0; IBM Corp., Armonk, NY, USA), with statistical significance defined as a *p*-value of less than 0.05.

## 3. Results

### 3.1. Baseline Characteristics

In total, 36 patients were recruited for the ICB00 group (mean age: 64.61 ± 6.34 years, range: 53–76 years); 14 patients were men, and 22 were women. For the ZCB00 group, 25 patients (mean age: 69.48 ± 6.96 years, range: 58–86 years) were recruited; 14 patients were men, and 11 were women. There was a significant difference in age, with the ZCB00 group having a significantly higher average age than the ICB00 group (*p* = 0.023). There was no significant difference in sex distribution (*p* = 0.187) between the groups.

Preoperatively, there were no significant differences in data except age between the two groups. The average CMT was 419.78 ± 65.43 μm in the ICB00 group and 437.91 ± 86.91 μm in the ZCB00 group (*p* = 0.624). The mean preoperative UDVA (logarithm of the minimum angle of resolution, logMAR) was 0.46 ± 0.33 in the ICB00 group and 0.53 ± 0.36 in the ZCB00 group (*p* = 0.407). Preoperative CDVA was 0.20 ± 0.14 in the ICB00 group and 0.26 ± 0.25 in the ZCB00 group (*p* = 0.694). Preoperative axial length on average was 23.72 ± 0.87 mm in the ICB00 group and 23.89 ± 1.10 mm in the ZCB00 group (*p* = 0.603). No significant differences were found in preoperative CMT, axial length, anterior chamber depth, and steep and flat keratometry between the two groups (Table 1).

### 3.2. Visual Outcomes

Postoperatively, the mean UDVA and CDVA of the ICB00 group were respectively 0.22 ± 0.17 and 0.09 ± 0.09, whereas those of the ZCB00 group were respectively 0.20 ± 0.17 and 0.08 ± 0.15. Significant differences were not observed in UDVA or CDVA between the two groups. For the ICB00 group, the mean UIVA measured at 80 and 60 cm were 0.33 ± 0.21 and 0.41 ± 0.21, respectively, while that of UNVA at 40 and 33 cm were 0.54 ± 0.17 and 0.46 ± 0.16, respectively. Similarly, for the ZCB00 group, the mean UIVA measured at 80 and 60 cm were 0.41 ± 0.19 and 0.50 ± 0.23, respectively, while that of UNVA at 40 and 33 cm were 0.54 ± 0.19 and 0.53 ± 0.23, respectively. Although the ICB00 group showed better UNVA at 33 cm distance and UIVA at 60 and 80 cm distance, statistically significant differences were not found in near or intermediate visual acuity. The average postoperative CMT was 407.69 ± 44.60 μm in the ICB00 group and 406.76 ± 52.87 μm in the ZCB00 group, with no significant difference between the two groups. The average postoperative SE was −0.15 ± 0.41 D in the ICB00 group and −0.18 ± 0.56 D in the ZCB00 group. The ME was 0.11 ± 0.54 D in the ICB00 group and 0.14 ± 0.39 D in the ZCB00 group. The MAE and MedAE were 0.42 D and 0.05 D in the ICB00 group, and 0.32 D and 0.21 D in the ZCB00 group, respectively. The postoperative SE and the prediction error showed no statistically significant differences between the two groups (Table 2).

### 3.3. Defocus Curve

In the defocus curve examination, both the ICB00 and ZCB00 groups demonstrated the best visual acuity outcomes at 0.0 D. The ICB00 group exhibited better defocus curves at far to near distance (+2.0 to −3.0 D) compared with the ZCB00 group. In terms of the depth of focus, which refers to the range of lens power from 0 to the maximum negative power, the ICB00 group demonstrated wider depth of focus than that of the ZCB00 group, over which the visual acuity on average was 0.3 logMAR or better (Figure 1).

### 3.4. Contrast Sensitivity

The ICB00 group revealed comparable results with the ZCB00 group under photopic, mesopic, and mesopic conditions with glares in the contrast sensitivity test. Patients in the ZCB00 group showed slightly higher contrast sensitivity at 3.4, 7.1, and 14.2 cpd than those in the ICB00 group under photopic conditions, but no significant differences were found between the two groups (*p* > 0.05). Patients in the ZCB00 group showed better outcomes at 3.4, 7.1, and 14.2 cpd than those in the ICB00 group under mesopic conditions with glare off, but without significant differences between the groups (*p* > 0.05). Under mesopic conditions with glare on, ZCB00 group patients had better contrast sensitivity at 3.4, 7.1, and 14.2 cpd than those in the ICB00 group patients, without significant differences between the groups (*p* > 0.05). Overall, no significant differences were found between the groups at any spatial frequency under each condition (Figure 2).

### 3.5. Optical Quality and Patient Satisfaction

The results of the questionnaire assessing postoperative dysphotopsia, including glare, halo, or starburst, revealed no significant differences between the groups (*p* > 0.05) (Figure 3). The ICB00 group reported lower spectacle dependence in near and intermediate, as well as in distant vision than the ZCB00 group, but there were no significant differences between the two groups (*p* > 0.05). In patients’ satisfaction of near, intermediate, and far vision, no significant differences were observed between the two groups (*p* > 0.05) (Figure 3).

## 4. Discussion

This study compared the visual outcomes of the ICB00 with those of the ZCB00 in patients with idiopathic ERM who underwent cataract surgery combined with PPV. The results revealed that patients with idiopathic ERM who underwent combined PPV and cataract surgery with the implantation of the ICB00 demonstrated good visual acuity in the far, intermediate, and near distances and a wider depth of focus compared to those with the ZCB00. These results were achieved without compromising contrast sensitivity or overall quality of vision.

The ICB00 is the recently invented monofocal IOL with characteristics that are almost equal to those of the ZCB00, except for the modified aspheric anterior surface and a 1.5 micron thickness variation with a 2 mm diameter in the optic [7]. While the ZCB00 maintained a perfectly circular profile within its central 2.5 mm region, the ICB00 had a central deviation of approximately 1.0 mm from an ideal circular shape. This deviation indicates a gradual and smooth altitudinal change that remains within a range of less than 4 µm, highlighting the changes in optic designs between the ZCB00 and the ICB00 [8]. This approach increases the depth of focus by altering higher-order aberrations while maintaining the structure of the standard monofocal IOLs, differing from multifocal IOLs that create multiple focal points at different distances through diffraction and refraction mechanisms [9]. These characteristics of the ICB00 enabled approximately a 0.5 D increase in power and enhanced the depth of focus, thereby providing better intermediate vision and comparable distant visual outcomes compared with the ZCB00 [7,10,11].

Although the ICB00 is thought to be similar to extended depth-of-focus IOLs in mechanism, as it increases the depth of focus without forming multiple focal points, several studies have referred to the ICB00 as a “monofocal IOL with enhanced intermediate function”, “enhanced monofocal IOL”, or “advanced monofocal IOL” [9]. The ICB00 is a safe and effective alternative to the conventional monofocal IOLs for cataract surgery, eliminating the need for specific patient selection criteria such as pupil size, personality of the patients, tolerance for dysphotopsia, or preferred reading distance [12]. Since the introduction of the ICB00, many studies performed globally have described the clinical outcomes of the ICB00 following cataract surgery. Gerd U. et al. reported that in a study involving Europeans, the ICB00 group showed better intermediate vision and similar distant vision, with a better defocus curve in the near and intermediate range (−0.5 D to −2.0 D) compared with the ZCB00 group after cataract surgery [13]. There were no significant differences in contrast sensitivity, patients’ satisfaction, and visual symptoms such as starbursts, glares, and halos compared to the ZCB00 group [13]. Additionally, a study reported by Choi et al. involving Korean patients showed that the ICB00 group provided better intermediate and near vision while maintaining comparable distant vision and contrast sensitivity compared to the ZCB00 group [14]. The ICB00 is known for retaining the advantages of a monofocal IOL while also providing the benefit of improved intermediate vision. Although multifocal IOLs are often associated with visual symptoms, such as glares or halos, ICB00 has demonstrated comparable visual quality to the monofocal IOLs in previous studies [13].

In addition to the studies performed on patients without any ocular disorders other than cataracts, there have also been reports on the clinical outcomes of cataract surgery in patients with ocular morbidities. A previous study by Nam et al. found that the ICB00 resulted in better intermediate vision, lower spectacle dependence, and higher patient satisfaction in patients with early glaucoma after cataract surgery [15]. In this study, the ICB00 group showed superior outcomes in UIVA and also demonstrated a better defocus curve in the intermediate defocus range (−1.0 D to −1.5 D), with similar visual quality and overall satisfaction compared to the ZCB00 group 3 months after cataract surgery [15]. Thus, the ICB00 may provide better intermediate vision without compromising subjective satisfaction in patients with ocular morbidities such as glaucoma.

Recently, some reports have demonstrated that the ICB00 had visual acuity results comparable to those of the ZCB00 without compromising contrast sensitivity or quality of vision in patients with various retinal disorders. These studies involved patients with retinal morbidities such as vitreous hemorrhage, retinal detachment, ERM, and macular hole who underwent combined PPV and cataract surgery [4,5]. However, despite these positive outcomes, there are still controversies about whether using premium IOLs is beneficial compared to using standard monofocal IOL in patients with retinal pathology. Especially in progressing conditions or advanced stages, there have been significant concerns to recommend the implantation of premium IOLs. In our study, we aimed to address these concerns by focusing on a specific patient group of IB or IC ERM. We simplified the target group by limiting the patients to those who underwent combined PPV and cataract surgery specifically for uncomplicated idiopathic ERM. By excluding those with a history of ophthalmic surgery or other ophthalmic conditions, we specifically focused on the plausibility and benefits of using the ICB00 in ERM vitrectomy. This approach allowed us to obtain a clearer understanding of the clinical outcomes of the ICB00 in patients with idiopathic ERM compared to the ZCB00. Recently, a small-scale study was published involving patients with ERM who underwent combined PPV and cataract surgery in Italy; some of them were implanted with the ICB00 (*n* = 11), while others were implanted with the ZCB00 (*n* = 12) [16]. The ICB00 group showed better intermediate vision (*p* < 0.001) and similar distant vision compared to the ZCB00 group at 6 months of follow-up [16]. Also, there were no significant differences in contrast sensitivity between the two groups [16]. In terms of patients’ satisfaction, the ICB00 group had more favorable results compared to the ZCB00 group (*p* < 0.001) [16]. While this study provided valuable insights, our study included a larger number of patients, offering a more robust data set and potentially more reliable conclusions.

In our study, the ICB00 group demonstrated comparable vision at near, intermediate, and far distances, as well as similar levels of dysphotopsia. There were no significant differences in spectacle dependence, contrast sensitivity, and overall patients’ satisfaction. Compared to previous studies where the ICB00 group showed better intermediate vision than the ZCB00 group in patients with or without ocular morbidities, our study did not find significant differences in UIVA at both 60 cm and 80 cm, as well as in spectacle dependence. However, the ICB00 group showed a better result compared to the ZCB00 group in the defocus curve. To the best of our knowledge, we believe that this is the first study to report the use of ICB00 in combined PPV and cataract surgery in Asian patients with idiopathic ERM.

Our study has several limitations. First, this study was a retrospective chart review of a relatively small number of patients performed over a short period (3–6 months) after surgery, and only Korean people were included. Also, we included only patients with ERM at specific stages, which limits the applicability of choosing IOLs for all ERM patients. Second, there are several factors that cannot be controlled that may affect IOL function, such as age, pupil size, endothelial cell count, etc. Third, our study included patients who underwent monocular surgery, but the questionnaire for subject satisfaction was conducted based on daily life experiences. The condition of the opposite eye may have affected the results of the questionnaire. Therefore, a prospective study with a larger sample size, enabling subgroup analyses over a longer term and controlling for detailed variables, is warranted.

## 5. Conclusions

In conclusion, the ICB00 showed good visual acuity and a wider defocus curve compared to the ZCB00 without decompensating contrast sensitivity, optical quality, or patient satisfaction in combined PPV and cataract surgery of patients with ERM requiring combined PPV and cataract surgery. Thus, ICB00 may be considered a safe and effective option for patients with idiopathic ERM.

## Figures and Tables

**Figure 1 bioengineering-11-00939-f001:**
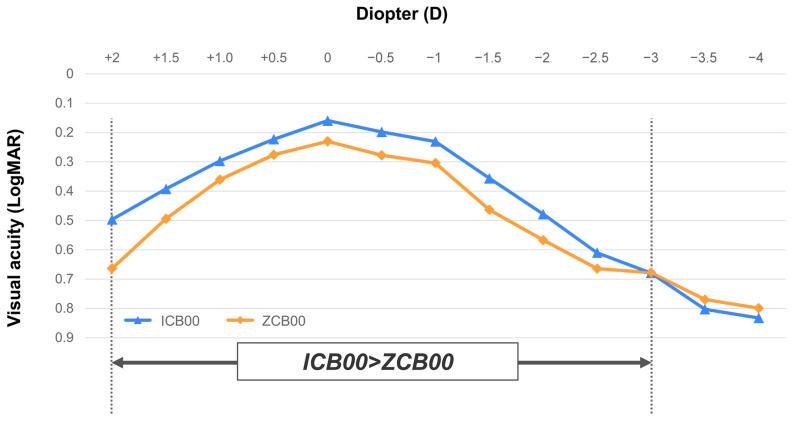
Defocus curves of the ICB00 and the ZCB00. Both groups exhibit the best visual acuity outcomes at 0.0 D, and the ICB00 group shows better defocus curves at far to near distances (+2.0 to −3.0 D). The ICB00 group demonstrates a wider depth of focus than that of the ZCB00 group, over which the average visual acuity is 0.3 logMAR or better. logMAR = logarithm of the minimal angle of resolution; D = diopters.

**Figure 2 bioengineering-11-00939-f002:**
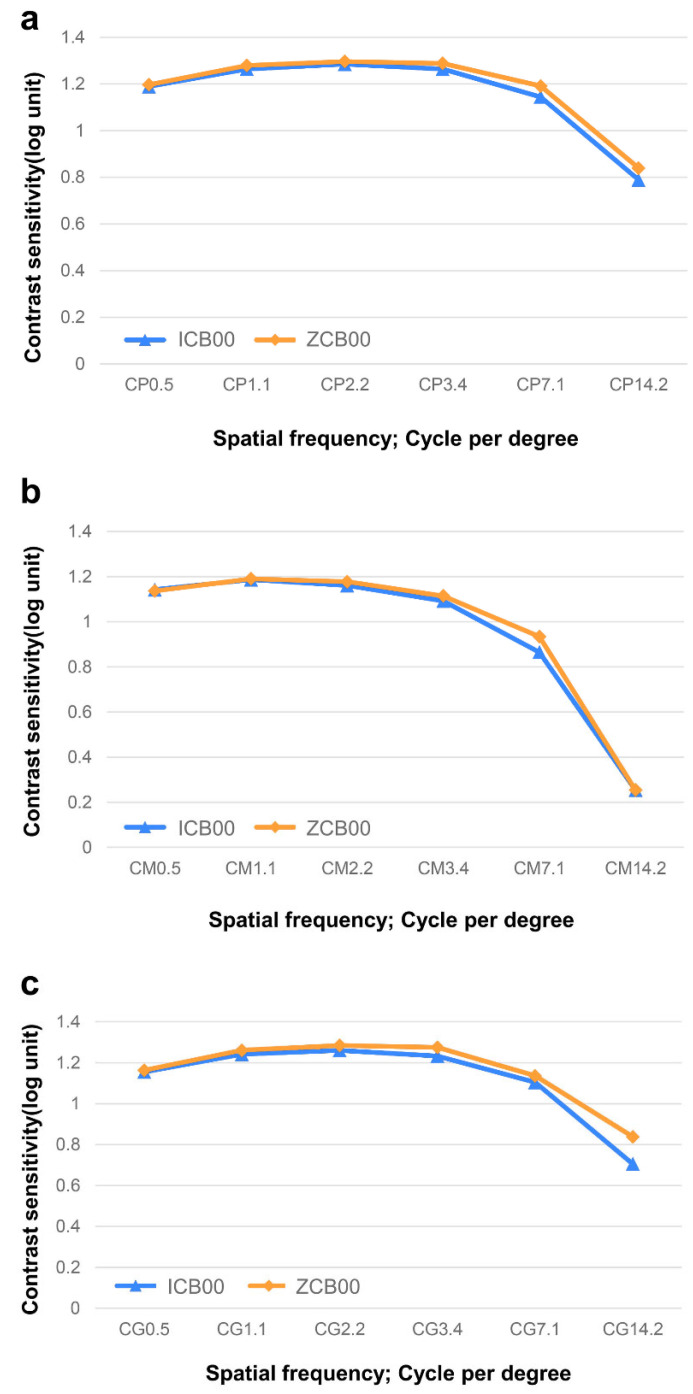
Contrast sensitivity of the ICB00 and the ZCB00. (**a**) Photopic. (**b**) Mesopic with glare off. (**c**) Mesopic with glare on. Both groups have normal values, without significant differences between the groups.

**Figure 3 bioengineering-11-00939-f003:**
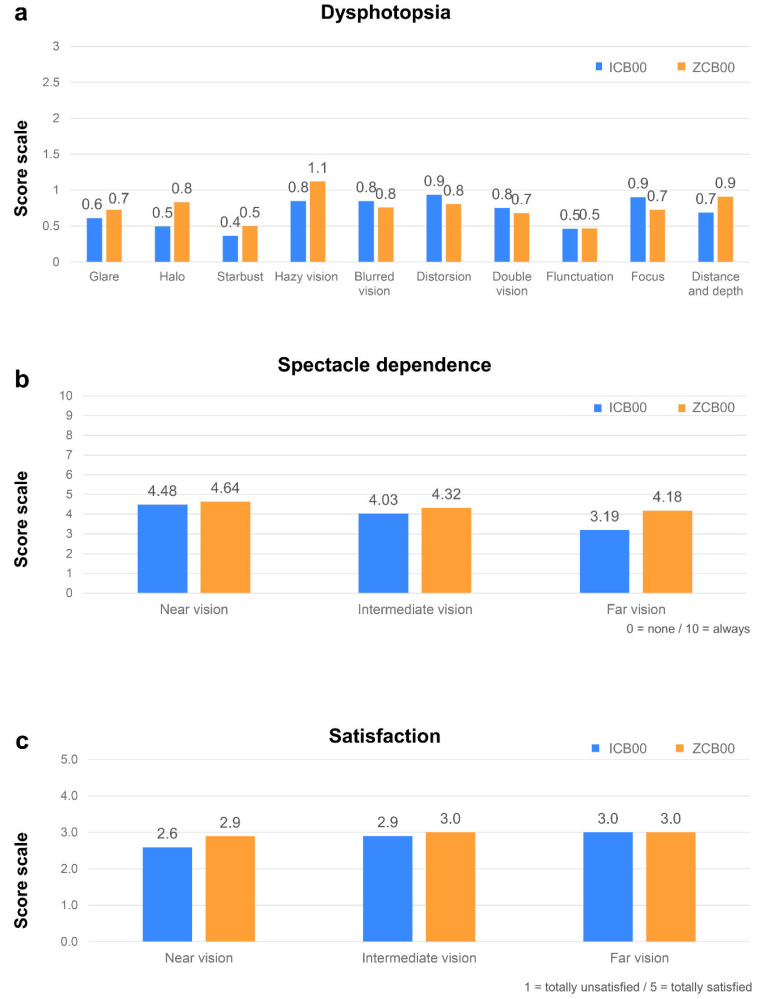
Postoperative questionnaire results at the 1-month follow-up. (**a**) Dysphotopsia questionnaire: There were no significant differences in the frequency of each symptom (glare, halo, starburst, hazy vision, blurred vision, distortion, double vision, fluctuation, focus, distance, and depth) between the two groups. Dysphotopsia score levels; 0 to 3 (none: 0; mild: 1; moderate: 2; severe: 3). (**b**) Spectacle dependence in the two groups on near and intermediate vision as well as on far vision: Both groups show relatively similar spectacle dependence in the visions of all distances, without significant differences in each distance. Spectacle dependence score levels; 0 to 10 (none: 0; N of 10: N; always: 10). (**c**) Satisfaction (near and intermediate, as well as far) in the two groups: No significant differences were observed in subjective satisfaction with all distances at the 1-month follow-up. Satisfaction score levels; 1 to 5 (totally dissatisfied: 1; dissatisfied: 2; neither satisfied nor dissatisfied: 3; satisfied: 4; totally satisfied: 5).

**Table 1 bioengineering-11-00939-t001:** Preoperative characteristics of the subjects.

Clinical Feature	ICB00 (N = 36)	ZCB00 (N = 25)	*p*-Value *
Age	64.61 ± 6.34 (53–76)	69.48 ± 6.96 (58–86)	**0.023**
Sex (M:F)	14:22	14:11	0.187
AXL	23.72 ± 0.87 (21.54–25.30)	23.89 ± 1.10 (22.13–25.73)	0.603
ACD	3.13 ± 0.42 (2.12–4.21)	3.04 ± 0.33 (2.42–3.65)	0.383
Steep K	44.68 ± 1.80 (42.03–49.67)	44.24 ± 1.39 (42.08–47.08)	0.416
Flat K	43.85 ± 1.60 (40.62–48.74)	43.37 ± 1.28 (41.56–45.73)	0.213
Preop UDVA (logMAR)	0.46 ± 0.33 (0.00–1.53)	0.53 ± 0.36 (0.10–1.70)	0.407
Preop CDVA (logMAR)	0.20 ± 0.14 (0.00–0.70)	0.26 ± 0.25 (0.00–1.00)	0.694
Preop CMT (μm)	419.78 ± 65.43 (254–537)	437.91 ± 86.91 (286–783)	0.624

Values are presented as mean ± standard deviation. Values of visual acuity are converted to logMAR. logMAR = logarithm of the minimum angle of resolution; AXL = axial length; ACD = anterior chamber depth; K = keratometry; UDVA = uncorrected distance visual acuity; CDVA = corrected distance visual acuity; CMT = central macular thickness. * Mann–Whitney *U*-test.

**Table 2 bioengineering-11-00939-t002:** Outcomes of patients at 3 to 6 months postoperative.

	ICB00 (N = 36)	ZCB00 (N = 25)	*p*-Value *
UDVA	0.22 ± 0.17 (0.00–0.52)	0.20 ± 0.17 (0.00–0.82)	0.620
CDVA	0.09 ± 0.09 (0.00–0.22)	0.08 ± 0.15 (0.00–0.70)	0.233
CMT (μm)	407.69 ± 44.60 (272–485)	406.76 ± 52.87 (301–500)	0.708
Target (Barrett, D)	−0.29	−0.29	1.000
SE (D)	−0.15 ± 0.41 (−0.75–+0.75)	−0.18 ± 0.56 (−2.00–+0.50)	0.277
ME (D)	0.11 ± 0.54	0.14 ± 0.39	
MAE (D)	0.42	0.32	0.24
MedAE (D)	0.05	0.21	
UIVA 80 cm	0.33 ± 0.21 (−0.10–+0.92)	0.41 ± 0.19 (+0.10–+0.70)	0.080
UIVA 60 cm	0.41 ± 0.21 (+0.01–+0.80)	0.50 ± 0.23 (+0.10–+1.10)	0.202
UNVA 40 cm	0.54 ± 0.17 (+0.30–+1.0)	0.54 ± 0.19 (+0.10–+0.80)	0.608
UNVA 33 cm	0.46 ± 0.16 (+0.19–+0.70)	0.53 ± 0.23 (0.00–+1.00)	0.206

Values are presented as mean ± standard deviation. Values of visual acuity are converted to logMAR. LogMAR = logarithm of the minimum angle of resolution; UDVA = uncorrected distance visual acuity; CDVA = corrected distance visual acuity; CMT = central macular thickness; D = diopters; SE = spherical equivalent; ME = mean error; MAE = mean absolute error; MedAE = median absolute error; UIVA = uncorrected intermediate visual acuity; UNVA = uncorrected near visual acuity. * Mann–Whitney *U*-test.

## Data Availability

The data reported in the study can be obtained from the corresponding author upon request.

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
