# Peer review of "Visual Outcomes and Patient Satisfaction of Enhanced Monofocal Intraocular Lens in Phacovitrectomy for Idiopathic Epiretinal Membrane"

_bioengineering, 2024, doi:10.3390/bioengineering11090939_

Round 1
Reviewer 1 Report
Comments and Suggestions for Authors
Normally previous studies showed better intermediate vision and lower spectacle dependence with ICB00 compared to ZCBOO, as expected. Since this study found the opposite, there should be a reason, maybe the success of PPV surgery? It is very hard to compare the IOL’s in such conditions that may have many different factors. For instance, usage of intraoperative tamponade might even cause different results and authors have not mentioned them at all.
Author Response
Thank you for taking the time to review this article. We are very grateful for your detailed comments.
Comment 1: Normally previous studies showed better intermediate vision and lower spectacle dependence with ICB00 compared to ZCBOO, as expected. Since this study found the opposite, there should be a reason, maybe the success of PPV surgery?
Response 1: Thank you for pointing out. Although there were no significant differences in postoperative visual acuity and spectacle dependence between the two groups, the ICB00 group showed better postoperative near and intermediate visual acuity as well as lower spectacle dependence than the ZCB00 group.
Comment 2: It is very hard to compare the IOL’s in such conditions that may have many different factors. For instance, usage of intraoperative tamponade might even cause different results and authors have not mentioned them at all.
Response 2: We deeply agree with your opinion that there are many factors which can influence the outcomes. For our analysis, we focused only on uneventful cases without complications during the surgery and follow-up period. None of the the patients underwent unexpected intraoperative procedures or experienced significant postoperative complications. Therefore, we have added an exclusion criterion to reflect this. However, we included only patients at certain stages of ERM, and more than 1 surgeon were involved in the surgery and follow-up. Thus, we have addressed these limitations in the discussion and have revised the manuscript, as follows.
- <In the revised manuscript, 72-74 line>
(The exclusion criteria were as follows:) cases with unexpected intraoperative events (e.g. retinal break or significant vitreous hemorrhage) or significant postoperative complications (e.g. secondary glaucoma or endophthalmitis)
- <In the revised manuscript, 222-226 line>
Also, we included only patients with ERM at specific stages, which limits the applicability of choosing IOLs for all ERM patients. Furthermore, surgery and postoperative follow-up were performed by two surgeons, which could have affected the statistical comparison of the outcomes. A prospective study with a larger sample size, enabling subgroup analyses over a longer term is warranted.
Reviewer 2 Report
Comments and Suggestions for Authors
In this very good paper authors present an evaluation of the performance, optical and in terms of patient satisfaction, of an new intraocular lenses (ICB00) implement on cataract surgery in patients with a particular problem, idiopathic epiretinal membrane (ERM), who requires pars plana vitrectomy. The research was correctly designed and the conclusions established in a sound ground. The results are also compared with those obtained with an older IOL of the same brand.
Author Response
Comment 1: In this very good paper authors present an evaluation of the performance, optical and in terms of patient satisfaction, of an new intraocular lenses (ICB00) implement on cataract surgery in patients with a particular problem, idiopathic epiretinal membrane (ERM), who requires pars plana vitrectomy. The research was correctly designed and the conclusions established in a sound ground. The results are also compared with those obtained with an older IOL of the same brand.
Response 1: We deeply appreciate your consideration of this article. Thank you for your comments.
Reviewer 3 Report
Comments and Suggestions for Authors
Very minor corrections are required as follows:
Line 25 should read – a smoother defocus curve compared to the ZCB00 group
Line 46 should read – and compare them with those of the ZCB00 group
Line 72 should read – using a three port 23 gauge…..
Line 81-82 should read – mainly using the Barrett Universal II formula
Line 165 should read – The results of the questionnaire
Line 168-169 should read – relatively similar between the two groups
Line 172 should read – There were no significant differences
Line 178 should read – No significant differences were observed
Line 187 – space after ZCB00
Line 194-195 should read – in patients with early glaucoma
Line 196 should read - in patients with…
Line 209 should read – was published reporting that the……
Line 211 should read – We believe this is the first study…..
Line 214 should read – chart review of a relatively small number….
Line 218 should read – good visual acuity and a wider defocus curve….
Comments on the Quality of English LanguageThe level of technical English is very high. I could only see very minor corrections required in the overall draft.
Author Response
Comment 1: The level of technical English is very high. I could only see very minor corrections required in the overall draft.
Response 1: Thank you very much for your detailed corrections. We have revised the manuscript, as follows.
- <In the revised manuscript, 25 line>
a smoother defocus curve compared to the ZCB00 group.
- <In the revised manuscript, 45-46 line>
and compare them with those of the ZCB00 group
- <In the revised manuscript, 75-76 line>
using a three port 23 gauge
- <In the revised manuscript, 84 line>
mainly using the Barrett Universal II formula.
- <In the revised manuscript, 171 line>
The results of the questionnaire
- <In the revised manuscript, 175 line>
relatively similar between the two groups
- <In the revised manuscript, 178 line>
There were no significant differences
- <In the revised manuscript, 184-185 line>
No significant differences were observed
- <In the revised manuscript, 193 line>
ZCB00 without
- <In the revised manuscript, 200-201 line>
in patients with early glaucoma
- <In the revised manuscript, 202 line>
in patients with
- <In the revised manuscript, 215 line>
was published reporting that the
- <In the revised manuscript, 217-218 line>
We believe this is the first study
- <In the revised manuscript, 220 line>
chart review of a relatively small number
- <In the revised manuscript, 228 line>
good visual acuity and a wider defocus curve
Round 2
Reviewer 1 Report
Comments and Suggestions for Authors
The authors still did not mention about the intraoperative tamponade they have used.
Author Response
Comment 1: The authors still did not mention about the intraoperative tamponade they have used.
Response 1: Thank you for pointing out. None of our patients underwent intraoperative tamponade, so we have revised the manuscript to clearly state this fact, as follows.
<In the revised manuscript, 72-75 line>
cases with unexpected intraoperative events which require additional procedures such as intraoperative tamponade (e.g. retinal break or significant vitreous hemorrhage) or significant postoperative complications (e.g. secondary glaucoma or endophthalmitis)